# Mesenchymal Tumor Management: Integrating Surgical and Non-Surgical Strategies in Different Clinical Scenarios

**DOI:** 10.3390/cancers16172965

**Published:** 2024-08-25

**Authors:** Laura Samà, Giorgia Amy Rodda, Laura Ruspi, Federico Sicoli, Vittoria D’Amato, Salvatore Lorenzo Renne, Alice Laffi, Davide Baldaccini, Elena Clerici, Pierina Navarria, Marta Scorsetti, Alexia Francesca Bertuzzi, Vittorio Lorenzo Quagliuolo, Ferdinando Carlo Maria Cananzi

**Affiliations:** 1Sarcoma, Melanoma and Rare Tumors Surgery Unit, IRCCS Humanitas Research Hospital, Rozzano, 20089 Milan, Italy; giorgia.rodda@humanitas.it (G.A.R.); laura.ruspi@humanitas.it (L.R.); federico.sicoli@humanitas.it (F.S.); vittoria.damato@humanitas.it (V.D.); vittorio.quagliuolo@cancercenter.humanitas.it (V.L.Q.); ferdinando.cananzi@hunimed.eu (F.C.M.C.); 2Department of Biomedical Sciences, Humanitas University, Pieve Emanuele, 20072 Milan, Italy; salvatore.renne@hunimed.eu (S.L.R.); marta.scorsetti@hunimed.eu (M.S.); 3Department of Pathology, IRCCS Humanitas Research Hospital, Rozzano, 20089 Milan, Italy; 4Medical Oncology and Hematology Unit, IRCCS Humanitas Research Hospital, Rozzano, 20089 Milan, Italy; alice.laffi@cancercenter.humanitas.it (A.L.); alexia.bertuzzi@cancercenter.humanitas.it (A.F.B.); 5Radiotherapy and Radiosurgery Department, IRCCS Humanitas Research Hospital, Rozzano, 20089 Milan, Italy; davide.baldaccini@cancercenter.humanitas.it (D.B.); elena.clerici@humanitas.it (E.C.); pierina.navarria@cancercenter.humanitas.it (P.N.)

**Keywords:** mesenchymal tumor, soft tissue sarcoma, retroperitoneal sarcoma, extremity sarcoma, GIST, desmoid-type fibromatosis, non-surgical treatment, multidisciplinary approach, personalized medicine

## Abstract

**Simple Summary:**

Soft tissue sarcomas (STSs) are rare cancers, making up less than 1% of all adult malignancies. Traditionally, the primary treatment for STS has been surgical resection. However, non-surgical approaches are becoming increasingly important in specific clinical situations. In this review, we explore the role of non-surgical treatments in managing STS, including their use as a bridge to surgery, as alternatives to surgery, for improving surgical outcomes, and for managing cases where surgery is not an option. Our findings highlight the effectiveness of these strategies in enhancing patient care and outcomes, providing a more personalized and less invasive approach to STS treatment. This review aims to advance the understanding and application of non-surgical methods in the treatment of soft tissue sarcomas.

**Abstract:**

Mesenchymal tumors originate from mesenchymal cells and can be either benign or malignant, such as bone, soft tissue, and visceral sarcomas. Surgery is a cornerstone treatment in the management of mesenchymal tumors, often requiring complex procedures performed in high-volume referral centers. However, the COVID-19 pandemic has highlighted this need for alternative non-surgical approaches due to limited access to surgical resources. This review explores the role of non-surgical treatments in different clinical scenarios: for improving surgical outcomes, as a bridge to surgery, as better alternatives to surgery, and for non-curative treatment when surgery is not feasible. We discuss the effectiveness of active surveillance, cryoablation, high-intensity focused ultrasound, and other ablative techniques in managing these tumors. Additionally, we examine the use of tyrosine kinase inhibitors in gastrointestinal stromal tumors and hypofractionated radiotherapy in soft tissue sarcomas. The Sarculator tool is highlighted for its role in stratifying high-risk sarcoma patients and personalizing treatment plans. While surgery remains the mainstay of treatment, integrating advanced non-surgical strategies can enhance therapeutic possibilities and patient care, especially in specific clinical settings with limitations. A multidisciplinary approach in referral centers is vital to determine the optimal treatment course for each patient.

## 1. Introduction

Mesenchymal tumors are a diverse group of neoplasms originating from connective tissue cells, which can develop in structures such as muscle, fat, bone, and fibrous tissue. These tumors can be classified into four categories based on their biological behavior: benign, intermediate (locally aggressive), intermediate (rarely metastasizing), and malignant, as defined by the WHO Classification of Tumors [1]. Benign tumors generally do not recur locally and are usually treated with complete local excision. Intermediate locally aggressive tumors, such as desmoid-type fibromatosis, frequently recur and require wide excision to ensure local control but rarely metastasize. For intermediate tumors with low metastasizing potential and malignant tumors with a substantial risk of distant metastasis, the mainstay treatment for these tumors is complete surgical resection, which involves removing the tumor along with a “cuff” of surrounding normal tissue to achieve clear margins. This surgical approach is often complex and technically demanding, especially for soft tissue sarcomas (STSs), where precise excision is crucial to minimize recurrence risks. Retroperitoneal sarcomas present an even greater challenge, frequently requiring multivisceral resections. Such intricate surgeries are best performed in high-volume referral centers, where a multidisciplinary team can develop and implement the optimal treatment plan tailored to each patient. The outbreak of the COVID-19 pandemic deeply affected elective oncological surgery, mainly due to the conversion of most of the healthcare resources into COVID-19 units. Furthermore, access to referral centers was significantly limited due to the strict movement restrictions imposed during the lockdown periods [2,3,4]. The urgent need for prioritization caused by this unprecedented healthcare setting gave us the input to review the different “non-surgical” approaches in mesenchymal tumors and to analyze different peculiar clinical settings. Therefore, we have described four distinct scenarios in which alternative therapeutic options should be considered: improvement in surgical results; a bridge to surgery; a better alternative to surgery; and non-curative treatment when surgery is not indicated or feasible.

For each of these scenarios, a variety of specific examples have been provided to illustrate the different approaches and treatments that can be utilized in clinical practice, offering a broad understanding of the potential options available (Figure 1).

## 2. Improvement of Surgical Results

With increasing evidence supporting the benefits of perioperative treatments for STS, both short-term and long-term outcomes for patients may see significant improvements following postponed surgery (Figure 2).

### 2.1. Gastrointestinal Stromal Tumors (GISTs)

The use of tyrosine kinase inhibitors (TKIs) has completely changed the course of GISTs [5]. Neoadjuvant Imatinib therapy is associated with tumor downstaging, organ preservation, and improved oncological outcomes. Currently, neoadjuvant Imatinib is used to downsize locally advanced, unresectable, or borderline resectable GISTs, when upfront surgery would be associated with high-morbidity or risk of positive resection margins. High-risk GISTs are defined as tumors with a high risk of recurrence according to tumor size, site, mitotic rate, and intraoperative tumor rupture [6]. Organ preservation is extremely important in cases of unfavorable locations in which demolitive resection could cause functional impairment such as the duodenum, gastroesophageal junction, and the rectum [7,8]. Rectal GISTs represent 5% of GISTs and are the third most common. Considering the anatomic and functional complexity of the pelvis, surgical treatment of a rectal GIST is challenging. The introduction of Imatinib in the 2000s changed rectal GIST management and outcome, allowing less demolitive surgical resection and reducing the local recurrence (LR) rate. Canvar et al. analyzed 47 primary rectal GISTs stratified by the availability period of Imatinib: 17 patients in the pre-Imatinib era and 30 patients in the Imatinib era. In the Imatinib era, only 3% of the patients underwent abdominoperineal resection or total pelvic exenteration compared with 59% in the pre-Imatinib era. The five-year overall survival (OS), disease-specific survival (DSS), and recurrence-free survival (RFS) were higher in the Imatinib era compared with the pre-Imatinib era (91, 100, and 82% versus 47, 61, and 44% respectively). Perioperative Imatinib treatment is also associated with lower LR, even with positive resection margins. Positive margins were found in only 31% of the patients who received neoadjuvant Imatinib compared to 71% of the patients who did not receive preoperative therapy. In the Imatinib era, no LRs were observed, compared to 26% at five years in the pre-Imatinib era [9]. Neoadjuvant therapy with Imatinib can also facilitate minimally invasive surgery by promoting tumor shrinking. A 2022 study found that the diameter of primary gastric GISTs in patients who received preoperative Imatinib, after a median follow-up period of 7.2 months, had decreased by 34%. Smaller tumor size at the time of surgery was associated with the successful completion of minimally invasive surgery [8].

### 2.2. Retroperitoneal Sarcoma (RPS)

Extended surgery remains the mainstay of RPS treatment. The use of preoperative radiotherapy (RT) and/or chemotherapy (CT) is not completely defined in RPS and pelvic sarcoma management [10]. RPS surgery is extremely resource demanding, considering that it usually requires multivisceral resection. It is reasonable to adopt neoadjuvant strategies with the aim of postponing surgery and improving oncological outcomes.

Currently, neoadjuvant RT is feasible for G1-G2 retroperitoneal liposarcoma. The STRASS multicenter randomized trial aimed to assess abdominal RFS in patients with RPS who underwent RT combined with surgery versus surgery alone. Between 2012 and 2017, 266 patients were enrolled in the trial: 128 patients had surgery alone and 133 patients had RT followed by surgery. Abdominal RFS at three years was similar in both groups: 58.7% (95% CI 49.5–66.7) in the surgery group and 60.4% (51.4–68.2) in the RT plus surgery group. The median abdominal RFS was 4.5 years in the RT plus surgery group and five years in the surgery-only group. In the RT plus surgery group, nineteen patients (14%) progressed during RT, and three of them developed distant metastases. When evaluating the liposarcoma subgroup, the abdominal recurrence-free interval at three years was 33.4% (95% CI 24.0–43.1) in the surgery group and 31.1% (22.1–40.5) in the RT plus surgery group (HR 0.91, 95% CI 0.58–1.42). This study has suggested that preoperative RT may not be the standard treatment in all RPS, but it can improve outcomes in liposarcoma. High-grade RPS could benefit from neoadjuvant CT. The STRASS-2 clinical trial, which is a phase III randomized trial on neoadjuvant CT in retroperitoneal and pelvic sarcomas, is currently ongoing to assess the impact of CT on DSS. The aim is to recruit 250 patients with high-risk dedifferentiated liposarcoma (DDLPS) and leiomyosarcoma (LMS) who will be randomized to undergo histologically tailored anthracycline-based neoadjuvant CT or upfront surgery [11].

### 2.3. Extremity Soft Tissue Sarcoma (ESTS)

For intermediate and high-grade ESTS, limb salvage surgery with RT is the recommended local treatment, which has allowed for the reduction in limb amputation [12,13]. According to NCCN [14] and ESMO [15] guidelines, perioperative RT is recommended to improve local control in settings wherein adequate margins are not achievable or for high-grade, deep-seated tumors or tumors of 5 cm or larger in size. Preoperative RT is preferred over postoperative RT due to lower rates of irreversible late toxicity [16] and to assess tumor RT sensitivity [17]. Traditionally, the dose of preoperative RT has been 50 Gy delivered in 25 fractions. Currently, hypofractionated RT (HFRT) regimens that can reduce the overall treatment period are being validated in clinical trials [13]. Conventional 5-week RT has little use in patients traveling long distances, reducing the accessibility to high-volume sarcoma centers. A recent single-center phase II clinical trial has evaluated the safety of a five-day equivalent of neoadjuvant RT. A cumulative dose of 30 Gy was administered in five fractions of six Gy each. Overall, 50 patients received treatment that was well tolerated, with no grade three or higher acute toxicity. At a median two-year follow-up, five (14.7% of 34 evaluated) developed grade ≥two fibrosis, lymphedema, and/or joint stiffness. Major wound complications occurred in 16 (32%) patients, which is in line with reported results after standard neoadjuvant RT. During the two-year period of the trial, the number of patients who received neoadjuvant RT increased by three folds, which suggests that a shorter treatment can increase accessibility to neoadjuvant therapy at high-volume centers [18]. A recently published survey collected management decision making and surveillance strategies of worldwide sarcoma specialists for patients with ESTS. Overall, 396 specialists responded. High-risk patients were defined based on histological subtype, grade, size, and tumor necrosis on magnetic resonance imaging (MRI). Regarding RT, of 301 responders, 47.2% treated high-risk patients with perioperative RT. In Asia, RT was offered less often (17.5%) than in Europe (52.1%; *p* < 0.001) or North America (62.4%; *p* < 0.001). The main features that influenced the use of RT were positive margins or expected positive margins [19]. Roohani et al. conducted a systematic review of the literature on the outcomes of preoperative HFRT in the STS of the extremity and trunk. Regarding oncological outcomes, HFRT showed positive results with a local control ranging between 80 and 100% from three to five years in the largest studies [19]. In a phase II trial in which 311 patients were treated with a preoperative RT scheme of five Gy in five fractions for a total dose of 25 Gy, an LR rate of 13.8% was achieved [20]. In the previously mentioned phase II trial by Kalbasi et al., at two-year follow-up, LR was observed in only 5.7% of the patients [18]. The DOREMY trial assessed the safety and oncological outcome of preoperative RT dose reduction in 79 patients with myxoid liposarcoma, a radiosensitive histological subtype. The reduced regimen was 36 Gy in two Gy fractions followed by surgery after a four-week interval. Extensive pathological response was found in 91% of the specimens. The OS rate was 95% at five years, with no local relapses at 2.1 years. Wound complications were observed in 22% of the patients, while grade two or higher acute toxicity was seen in 14% of the cases [21]. According to these results, preoperative HFRT can be safely used in ESTS. Especially, in distinct settings, such as the COVID-19 pandemic, the employment of HFRT regimens demonstrated considerable advantages by reducing the total duration of treatment and enhancing patient adherence [22].

In advanced and metastatic disease, CT is the mainstay treatment [15]. In localized disease, the administration of CT is less defined. According to the previously mentioned international survey, no significant differences have emerged in the use of CT in high-risk sarcoma among the continents, and the main factors that influenced the use of perioperative CT were histological subtype, grade, size, tumor differentiation, performance status, and age [19]. When neoadjuvant CT is proposed, the standard regimen is anthracycline-based CT, which is the first-line therapy for unresectable disease [23]. A prospective randomized phase III trial has evaluated if histotype-tailored CT is superior to standard neoadjuvant anthracycline plus ifosfamide in five high-risk subtypes of the STS of the extremities and trunk: myxoid liposarcoma (HG-MLPS), LMS, synovial sarcoma (SS), malignant peripheral nerve sheath tumor (MPNST), or undifferentiated pleomorphic sarcoma (UPS). High-risk criteria were grade three and a diameter equal to or greater than 5 cm. Overall, 287 patients were included, 145 in the standard CT group and 143 in the histotype-tailored group. No superiority in terms of disease-free survival (DFS) and OS was found for histotype-tailored CT over standard CT, suggesting that anthracycline-based CT should remain the preferred regimen in high-risk STS [24]. Although there is no prospective evidence in favor of the use of neoadjuvant CT, the use of the Sarculator, a risk stratification tool, has further refined the approach to neoadjuvant CT in STS. By calculating individualized risk based on factors like age, tumor size, and histology, the Sarculator aids in identifying patients who would most benefit from intensive treatments, such as anthracycline-based CT. This personalized risk assessment ensures that high-risk patients receive the most appropriate and potentially effective treatment, improving overall outcomes [25].

In the last few years, the use of checkpoint inhibitors such as ipilimumab (anti-CTLA4), pembrolizumab (anti-PD1), and nivolumab (anti-PD1) has improved the treatment of certain tumors, like melanoma and non-small-cell lung carcinoma [26]. While STS is considered immunologically quiet compared to other solid tumors with high mutation rates [27], some histological subtypes have complex genomic features and an immune infiltration, which may represent a target for checkpoint inhibitors. The SARC028 phase II trial has demonstrated promising activity of anti-PD1 pembrolizumab on UPS and DDLPS [28]. Furthermore, RT can improve the effect of immunotherapy by increasing antigenic expression and releasing inflammatory cytokines that induce the tumor expression of death receptors [18]. A randomized phase II trial by the MD Anderson Cancer Center tested the effects of neoadjuvant immune checkpoint blockade (ICB) therapy in patients with resectable retroperitoneal DDLPS and extremity/truncal UPS. It explored the use of nivolumab alone or combined with ipilimumab, with some patients also receiving RT. The study found significant pathologic responses in UPS with concurrent ICB and RT, while the response in DDLPS was less robust. The presence of intratumoral B cells and lower densities of regulatory T cells were associated with better outcomes [29].

## 3. Bridge to Surgery

There are specific cases where a non-surgical approach can represent a viable alternative or, at the least, serve as a crucial bridge to surgery for the treatment of tumors, providing patients with additional options and time to prepare for the surgical procedure (Figure 3).

### 3.1. Atypical Lipomatous Tumors (ALTs)

Extremity and trunk well-differentiated liposarcomas, which are more accurately classified as ALTs, are usually intermediate tumors with very little metastatic potential [30]. These tumors are mostly treated surgically, which can potentially induce severe morbidities. In a recent retrospective study, Vos and colleagues evaluated the outcomes of patients with ALTs who underwent either active surveillance or surgery. The study found that 5 out of 191 surgically treated patients developed metastatic disease (2.6%); notably, 3 of these patients developed metastases following local dedifferentiated recurrence. The five-year disease survival rate for these patients was remarkably high at 98.5%. Active surveillance was conducted on 24 patients for a median period of 1.8 years. Out of these twenty-four patients, four eventually required surgery due to the onset of symptoms or tumor growth. The evaluation of ALT natural history indicated that patients rarely die unless dedifferentiation occurs, with low rates of dedifferentiation reported in the literature [31,32]. Given these observations, active surveillance could be an appropriate strategy to pursue in selected patients who do not experience any symptoms, thereby avoiding unnecessary surgical interventions and associated morbidities [30].

### 3.2. GIST

Small GISTs represent another case in which active surveillance could be an adequate alternative or a temporary bridge to surgery. The standard approach for managing esophagogastric and duodenal GISTs smaller than 2 cm is surgical resection or, if it can be performed without causing tumor rupture, endoscopic resection. According to the 2022 ESMO Guidelines, an option for these small-sized GISTs is to undergo active surveillance, with the decision depending on factors such as tumor origin site, patient age, life expectancy, and comorbidities [5]. The most recent NCCN guidelines recommend the resection of small GISTs in symptomatic patients or if endoscopic ultrasound (EUS) reveals high-risk features, such as irregular borders, echogenic foci, or sonographic heterogeneity [14]. In a recent study, Patel and colleagues evaluated the safety of expectant management (EM) in patients with small GISTs. The study included 1330 patients from the National Cancer Database (NCDB). The findings showed no significant difference in five-year OS between patients undergoing EM and those undergoing surgery (95.7% vs. 92.6%, *p* = 0.4882) [33]. Currently, no follow-up protocol is validated for the active surveillance of GISTs. A logical approach is to perform a three-month evaluation followed by increasing the follow-up interval if the tumor remains stable. Within a prioritization system, surgical excision could be reserved for patients with small-sized GISTs whose tumors increase in size or become symptomatic over time [33]. For intermediate or high-risk GISTs that carry Imatinib-sensitive mutations, if access to surgical treatment is limited, Imatinib may be a viable option to avoid tumor progression and manage the disease effectively. This strategy allows for a more flexible and patient-tailored approach to treating small GISTs, ensuring that each patient receives the most appropriate care based on their individual circumstances.

### 3.3. Dermatofibrosarcoma Protuberans (DFSPs)

Classical DFSP, which accounts for 90% of cases, typically has an indolent course characterized by a locally invading growth pattern but a low potential for metastasis. In contrast, the high-grade fibrosarcomatous variant (FS-DFSP), which makes up about 10% of cases, carries a significantly higher risk of metastasis [34]. Although the mainstay treatment for DFSP is complete surgical resection, in settings with limited resources, considering the characteristics of DFSP, a watchful waiting approach might be considered. Due to the typical superficial spread of DFSP, if the resection is not correctly planned, it can result in an incomplete excision with microscopic positive margins, known as an R1 resection. A study published in 2019 demonstrated that fibrosarcomatous change and positive resection margins were independent prognostic factors for LR [35]. In view of these results, following an R1 resection, a second excision to achieve clear margins might be indicated, even though it could lead to extensive soft tissue defects that require plastic reconstruction. In a specific setting where surgical procedures are limited, clinical follow-up could represent a valid alternative to surgical re-excision after an R1 resection [36], leaving a second surgical excision only for cases of LR. Additionally, when surgery is not available, and the patient presents with FS-DFSP or a rapidly growing tumor, Imatinib could be considered as an alternative treatment to control tumor progression. This approach allows for the management of DFSP in a more flexible manner, ensuring that each patient receives appropriate care based on the resources available and the specific characteristics of their condition. By employing a watchful waiting strategy or medical therapy, like Imatinib, in certain cases, healthcare providers can optimize patient outcomes, even in constrained clinical environments.

## 4. Better Alternative to Surgery

In well-selected cases, where the biological characteristics of specific tumors suggest a less aggressive behavior, a watchful waiting approach could represent a better alternative to immediate surgical resection. This method involves regular monitoring of the tumor through follow-up visits and imaging studies to ensure that any changes in the tumor’s behavior are promptly detected and addressed. In this way, a watchful waiting approach can provide a more patient-centric treatment plan, allowing for interventions only when necessary (Figure 4).

### 4.1. Desmoid-Type Fibromatosis (DTF)

The outcome of DTF is unpredictable given that spontaneous regression, long stable disease, or progression could occur. Currently, a watchful waiting approach tends to replace large en bloc resection as the first therapeutic approach since recurrence after surgical excision develops in over 60% of cases, and spontaneous regression is seen in 25% of cases [37]. A 2017 prospective study conducted by the French Sarcoma Group analyzed outcomes and prognostic factors of 771 confirmed cases of DTF regarding initial patient management. Event-free survival (EFS) at two years was 56%, with no significant difference between patients who underwent surgery and those managed with a watchful waiting approach (53% versus 58%, *p* = 0.415). In univariate analysis, primary location was an independent prognostic factor. The two-year EFS was 66% for favorable locations (abdominal wall, intra-abdominal, breast, digestive viscera, and lower limb) and 41% for unfavorable locations. Among patients with favorable locations, the two-year EFS was similar in patients treated by both surgery (70%) and the watchful waiting approach (63%; *p* = 0.413). Among patients with unfavorable locations, the two-year EFS was significantly higher in patients initially managed with a watchful waiting approach (52%) compared with those who underwent upfront surgery (25%; *p* = 0.001) [38]. According to these results, a watchful waiting approach should be preferred in the initial management of DTF in unfavorable locations. Since there is a possibility of disease progression, a long follow-up is crucial. A more recent study compared management strategies, upfront surgery vs. watchful waiting, in 87 consecutive patients diagnosed with DTF in two different periods: the early period (2000–2010) and the late period (2012–2018). In the early period, upfront surgery was performed in 94.4% of cases, while in the late period, it was performed in 27.3% of cases. No statistically significant difference was found in EFS between the two groups. This study reflected the switch of paradigm in DTF management, recently tending to a wait-and-see approach [39].

Consistently, recent findings from European prospective observational studies on DTF have shown that active surveillance, rather than immediate aggressive treatment, is effective for newly diagnosed DTF patients. The findings validate previous retrospective data and align with the 2020 consensus guidelines advocating for active surveillance as the initial approach [40,41,42]. In conclusion, according to the 2023 update to the global evidence-based consensus guideline, after a biopsy-confirmed diagnosis of DTF, active surveillance should represent the first-line approach [43]. After active surveillance failure, if DTF requires treatment, non-surgical approaches should be considered.

Cryoablation is an effective alternative treatment, especially for small- to moderately-sized extra-abdominal DTF. The CRYODESMO-01 trial, a prospective, multicenter trial, assessed cryoablation in non-abdominopelvic progressive DTF, demonstrating an 86% non-progression rate at 12 months, reduced pain, and improved functional status. Tumor size was the only variable associated with treatment failure, with progression observed at the periphery of the cryoablated zone, suggesting that cryoablation could be a definitive treatment for smaller DTFs [44]. The current indication for using cryoablation is a growing DTF after two or more lines of medical therapy or with functional symptoms or pain [43].

High-intensity focused ultrasound (HIFU) is a thermal ablation technique used for superficial, small-sized DTFs [45]. However, due to the retrospective and non-randomized study nature [46,47,48] and the risk of neural or skin injuries and post-procedure edema, HIFU is not considered a primary treatment option.

Techniques like radiofrequency, microwave ablation, chemical ablation, and transarterial chemoembolization have shown success in small retrospective series and case reports but require more prospective and long-term outcome data [43].

### 4.2. Schwannoma

Schwannomas are rare, usually benign nerve sheath tumors. When they occur intrabdominally, retroperitoneally, or within the pelvic region, they pose significant management challenges due to their complex locations and potential for impacting surrounding structures. A study published in 2020 by the Transatlantic Australasian Retroperitoneal Sarcoma Working Group (TARPSWG) established evidence-based recommendations for the management of abdominal, retroperitoneal, and pelvic schwannomas. According to this comprehensive work, early indications for surgical resection include the presence of symptoms caused by the tumor, diagnostic uncertainty that cannot be resolved through non-invasive means, and evidence of rapid tumor expansion. In cases where there is no immediate indication for surgery, it is essential to confirm the diagnosis through a biopsy. Following this, the patient should undergo regular radiological follow-up for at least two years to better understand the tumor’s behavior and monitor any changes. This approach ensures that any necessary interventions are timely and appropriate while avoiding unnecessary surgery when it is not indicated [49].

### 4.3. Ganglioneuroma

Ganglioneuromas frequently present asymptomatically, meaning that patients often do not experience any noticeable symptoms, and these tumors tend to have an indolent growth pattern, growing very slowly over time. The disease course and clinical management of ganglioneuroma were evaluated in a comprehensive retrospective study conducted by TARPSWG, which assessed the role of active surveillance. Overall, the study included 328 patients from 29 institutions who presented with primary ganglioneuroma between 2000 and 2020. Of these patients, 116 (35.4%) underwent active surveillance, while resection was carried out in 212 (64.6%) patients, primarily due to symptomatic disease, diagnostic uncertainty, or large tumor size. The patients who underwent active surveillance had a median follow-up period of 1.9 years, and 92.2% of ganglioneuromas remained stable in size during this time. This study revealed that the risks associated with surgery often exceed the benefits for patients with asymptomatic or indolent tumors. Consequently, in these selected cases, active surveillance may be more suitable and preferable to surgical intervention, allowing patients to avoid the potential complications and risks associated with surgery [49].

## 5. Non-Curative Treatment

Surgical resection of mesenchymal tumors is not always feasible due to patient characteristics or, in the case of sarcomas, disease progression. In such instances, alternative techniques can be employed to limit tumor growth, reduce symptoms, and improve patient outcomes (Figure 5).

The management of unresectable or metastatic STS remains a primary concern. In a metastatic setting, anthracycline-based CT has the lead role, but RT can be delivered to improve tumor control, especially if access to surgery is limited. Given the relative resistance of large-sized sarcomas to conventional RT and the anatomic limitations of stereotactic body RT (SBRT), HFRT is emerging for patients with unresectable or metastatic sarcoma. A one-year DSS of 59% after HFRT is reported, which is higher in patients with oligometastatic (100%) or oligoprogressive disease (73%). HFRT can offer a one-year local control for targeted lesions of 73% and can provide palliative relief in 95% of the cases. HFRT is an effective treatment option in oligometastatic patients, as it offers durable local control, symptom palliation, and CT breaks with limited toxic effects [50].

Additionally, spatially fractionated radiation therapies, specifically GRID and Lattice techniques, represent advanced approaches for treating radioresistant and large-volume STS. These methods deliver high doses directly to the tumor while minimizing damage to healthy tissues, enhancing tumor control and reducing treatment-related toxicity. Combining GRID and Lattice therapy with CT, immunotherapy, or conventional external beam radiation therapy (EBRT) can further improve outcomes. Despite planning challenges, these techniques offer hope for difficult-to-treat cancers. Clinical trials and standardized guidelines will optimize these strategies, ensuring safe and effective treatment for bulky tumors [51]. Radiofrequency ablation (RFA) is another technique that can be used in metastatic STS. This treatment option is commonly used in unresectable non-small-cell lung cancer and lung colorectal cancer metastases.

Approximately 20–30% of patients with sarcoma present lung metastases. In this setting, RT combined with systemic CT does not greatly improve patient prognosis. The current recommended treatment for oligometastatic resectable pulmonary sarcoma metastases is aggressive surgical resection [15], but metastasectomy is not always feasible due to a context of limited resources or due to the patient’s comorbidities, disease multifocality, and early recurrence of resected metastases. Lung RFA is a less invasive technique that has been reported to have a three-year survival rate comparable with surgical metastasectomy (59% vs. 54%). The most important prognostic factor is the complete ablation of lung metastases with a one-year survival rate of 88% in patients who underwent complete ablation versus 29% in cases of incomplete ablation [52].

Percutaneous thermal ablation (PTA) is a noninvasive local treatment that is gaining recognition as a well-tolerated therapy for primary and metastatic small lesions in the liver, lung, and bones. PTA for sarcomatous metastases has been demonstrated to provide effective local control and to improve OS in patients with oligometastatic disease, defined as patients with less than five lesions, regardless of histology and the site of the lesions (median 45.3 versus 12.6 months, adjusted hazard ratio = 0.47, *p* < 0.001) [53]. Gravel et al. have evaluated the use of PTA in metastatic LMS, either pulmonary or extrapulmonary. The median OS was 48.3 months and the one-, three-, and five-year OS rates were 96.7%, 62.0%, and 28.3%, while the local control rate at one year was 95.2%. PTA represents an effective local treatment option for local metastatic control for pulmonary and extrapulmonary disease, especially in those patients unfit for surgery [54].

Non-surgical approaches can also play a role in tumor complications. A rare but clinically relevant complication is gastrointestinal bleeding from GIST. It usually manifests as asymptomatic occult bleeding but in rare cases, it can present as life-threatening bleeding. Endoscopic intervention is the treatment of choice, but in case of failure, emergency surgery is necessary, although it carries significant morbidity and mortality. Transcatheter arterial embolization (TAE) can be an effective alternative to surgery in the management of gastrointestinal bleeding from GIST, especially in elderly patients. A recent study reported a series of 20 patients who underwent TAE for gastrointestinal bleeding from GIST. Of these, 17 patients (85%) presented to the emergency department with acute, unstable hemodynamics showing hematochezia, melena, hematemesis, hemoperitoneum, and sudden hypotension, and 14 patients had undergone endoscopy. Technical and clinical success rates of TAE were 95% (19 of 20 patients) and 90% (18 of 20 patients), with no TAE-related complications, such as bowel ischemia [55].

## 6. Conclusions

This review has emphasized the critical role of non-surgical treatments in the management of mesenchymal tumors, demonstrating their utility not only as independent therapies but also as integral components of multimodal treatment strategies. Non-surgical options are especially crucial in scenarios where surgery poses high risks or is not feasible, highlighting a shift towards more personalized and less invasive management approaches.

Historically, non-surgical treatments were underutilized due to a lack of understanding and technological constraints. However, recent advances have significantly changed this landscape. For instance, the development of tyrosine kinase inhibitors for GIST and the application of HFRT have markedly improved survival rates and quality of life for patients. Additionally, immunotherapy, particularly immune checkpoint inhibitors, such as pembrolizumab and nivolumab, has shown promising results in enhancing the body’s immune response against certain sarcoma subtypes.

Future research should prioritize refining these techniques, integrating advanced technologies, and developing novel agents to tailor treatments more closely to individual patient profiles. By continually advancing non-surgical strategies and incorporating them into clinical practice, we can expand therapeutic possibilities and achieve more effective, personalized patient care.

In conclusion, the strategic integration of non-surgical treatments into the management of mesenchymal tumors, particularly in specific clinical scenarios, offers valuable alternatives to traditional approaches. These strategies can enhance the overall management of these tumors and highlight the crucial role of multidisciplinary collaboration in optimizing patient care.

## Figures and Tables

**Figure 1 cancers-16-02965-f001:**
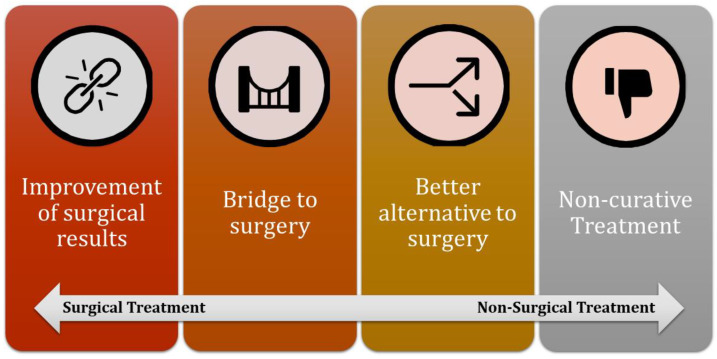
Different clinical scenarios of non-surgical management of soft tissue sarcomas.

**Figure 2 cancers-16-02965-f002:**
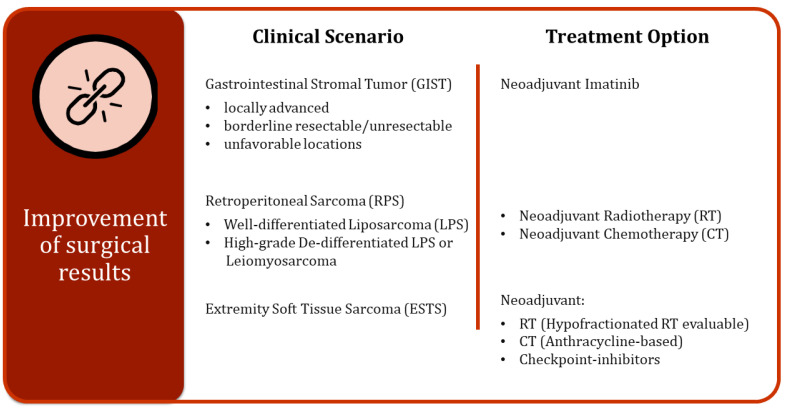
Different clinical scenarios and related treatment options for improving surgical results.

**Figure 3 cancers-16-02965-f003:**
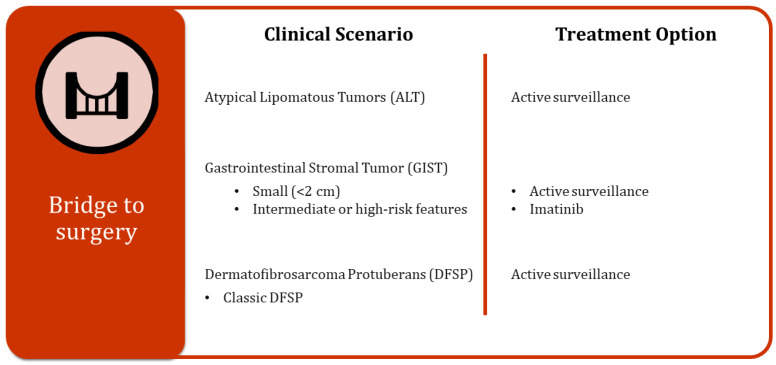
Different clinical scenarios and related treatment options as a bridge to surgery.

**Figure 4 cancers-16-02965-f004:**
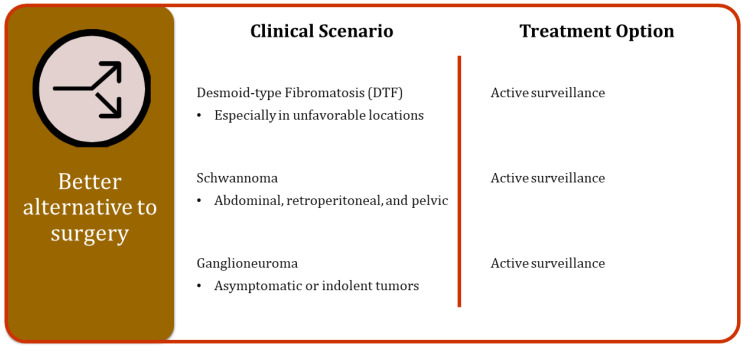
Different clinical scenarios and related treatment options as alternatives to surgery.

**Figure 5 cancers-16-02965-f005:**
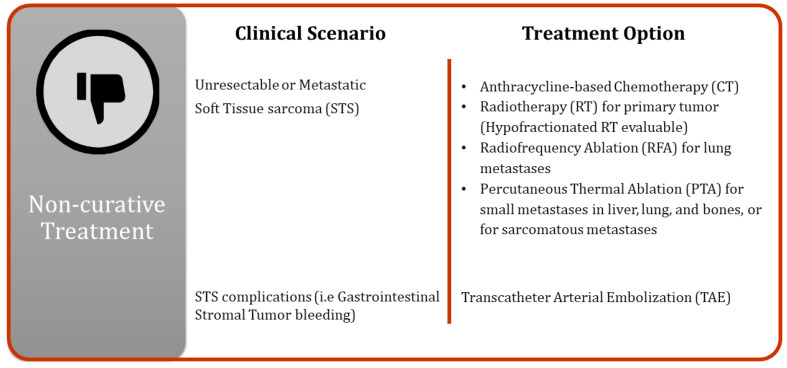
Non-curative treatment approaches.

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
