# Peer review of "Mesenchymal Tumor Management: Integrating Surgical and Non-Surgical Strategies in Different Clinical Scenarios"

_cancers, 2024, doi:10.3390/cancers16172965_

Round 1

Reviewer 1 Report (Previous Reviewer 2)

Comments and Suggestions for Authors

The Authors described the role of non-surgical treatments in managing STS 

The topic is interesting, but the review is disorganized and confusing. Mainly, I think that an analysis of different scenarios rather than describing different histologies would be clearer. At present, figures are unuseful and confusing.

Other entities such as desmoid fibromatosis are wrongly considered as tumours.

Moreover, would further specify on which are possible alternatives to surgery.

Finally, the conclusions are not supported by results.

Comments on the Quality of English Language

Many grammar and syntax errors. 

Author Response

Comments 1: The Authors described the role of non-surgical treatments in managing STS. The topic is interesting, but the review is disorganized and confusing. Mainly, I think that an analysis of different scenarios rather than describing different histologies would be clearer.

Response 1: Thank you for your comments. We appreciate your feedback and understand the importance of clear organization in the review. Given the diverse scenarios in managing mesenchymal tumors, we found it challenging to cover all aspects comprehensively. We chose to structure the review around histologic subtypes within each clinical scenario to provide clarity and avoid confusion. This approach was intended to balance scenario-based analysis with the specific challenges posed by different histologies. We value your suggestion and will keep it in mind for future works.

Comments 2: At present, figures are unuseful and confusing.

Response 2: Thank you for your feedback. We sincerely believe that the figures are an important tool for visually representing key concepts, enhancing the reader's comprehension of the content. Our intention was to offer a clear and concise visual summary of the information discussed, and we hope these figures assist the reader in navigating the manuscript more effectively.

Comments 3: Other entities such as desmoid fibromatosis are wrongly considered as tumours. Moreover, would further specify on which are possible alternatives to surgery.

Response 3: Thank you for your comment. Desmoid fibromatosis is recognized as a tumor of uncertain behavior according to the 2020 WHO classification, and terms like "aggressive fibromatosis" and "desmoid tumor" are synonymous. We included it in our review based on this classification. Additionally, we have detailed the possible alternative strategies to surgery in Paragraph 4.1 of the manuscript. We hope this clarifies the inclusion and addresses your concerns.

Comments 4: Finally, the conclusions are not supported by results.

Response 4: We would like to clarify that the conclusions in our manuscript were intended to provide a concise summary of the key points discussed in this review. Given the nature of a review article, the conclusions are meant to encapsulate the overarching insights derived from the literature rather than present new data or results. However, we have taken your comments into consideration and made some adjustments to the conclusions to better reflect the discussions and findings within the manuscript.

Comments on the Quality of English Language: Many grammar and syntax errors. 

Response: In light of your comment regarding the quality of the English language, we have taken the necessary steps to address this issue. Specifically, one of the co-authors (GR), who is a native English speaker, has thoroughly reviewed the entire manuscript and made the necessary grammatical and syntactical corrections, which can be found in the attached manuscript.

Reviewer 2 Report (New Reviewer)

Comments and Suggestions for Authors

I can very well imagine why two reviewers are against and other 2 pro. Too often one interprets only from someone’s own perspective… meaning a surgeon who can’t stand not to operate, and an radiation oncologist who simply does not want to use hypofractionation because the reimbursement is dramatically less. I do explain this situation -again and again- in this very context, as well as here too.

It is totally clear that although surgery is the absolute mainstay in sarcoma treatment, this is and shall never be an ideology, the sarcoma community has to strive for transdisciplinarity, not only in words but specifically in daily practice. It is very clear that nowadays it is not enough to think only from one mono-disicipline. It’s not state of the art any more, and we have to support any multidisciplinary thinking anyway. We all have to learn that choosing the “right” treatment options becomes more and more of a challenge in the armamentarium of the sarcoma expert, and the more experienced someone gets, the more difficult it gets to choose the best option. Simply and just because of these circumstances, this paper has to be accepted, because from this perspective, it great adds to the literature or fills a void!

Personally, I think the title might slightly be adapted to respect this better (something like: "Tailoring Mesenchymal Tumor Treatment: Integrating Non-Surgical and Surgical Approaches Based on Individual Patient Profiles"). 

Author Response

Comments 1: 

I can very well imagine why two reviewers are against and other 2 pro. Too often one interprets only from someone’s own perspective… meaning a surgeon who can’t stand not to operate, and an radiation oncologist who simply does not want to use hypofractionation because the reimbursement is dramatically less. I do explain this situation -again and again- in this very context, as well as here too.

It is totally clear that although surgery is the absolute mainstay in sarcoma treatment, this is and shall never be an ideology, the sarcoma community has to strive for transdisciplinarity, not only in words but specifically in daily practice. It is very clear that nowadays it is not enough to think only from one mono-disicipline. It’s not state of the art any more, and we have to support any multidisciplinary thinking anyway. We all have to learn that choosing the “right” treatment options becomes more and more of a challenge in the armamentarium of the sarcoma expert, and the more experienced someone gets, the more difficult it gets to choose the best option. Simply and just because of these circumstances, this paper has to be accepted, because from this perspective, it great adds to the literature or fills a void!

Responce 1: Dear Reviewer, thank you very much for your thoughtful and supportive feedback. We appreciate your recognition of the importance of a multidisciplinary approach in the treatment of sarcomas and other mesenchymal tumors.

Comments 2: Personally, I think the title might slightly be adapted to respect this better (something like: "Tailoring Mesenchymal Tumor Treatment: Integrating Non-Surgical and Surgical Approaches Based on Individual Patient Profiles"). 

Responce 2: Your suggestion to refine the title to better reflect the integration of surgical and non-surgical approaches is well noted. In line with your advice, we have revised the manuscript's title to: "Mesenchymal tumor management: integrating surgical and non-surgical strategies in different clinical scenarios"

This manuscript is a resubmission of an earlier submission. The following is a list of the peer review reports and author responses from that submission.

Round 1

Reviewer 1 Report

Comments and Suggestions for Authors

This review clearly and completely exposes the possible scenarios in which surgical treatment could be delayed, omitted or replaced, a truly very interesting and useful topic in daily clinical practice. 

I have no particular notes or suggestions except: 

- in paragraph 3.1 the authors should briefly mention the emerging role of cryoablation as a possible alternative treatment to surgery. 

- in paragraph 4.3 the authors should specify that, although there is no prospective evidence in favor of the use of neoadjuvant chemotherapy, the stratification of high-risk patients according to validated nomograms such as the "sarculator" has demonstrated a benefit from the use of treatments based of anthracyclines (Pasquali S, et al; Neoadjuvant chemotherapy in high-risk soft tissue sarcomas: A Sarculator-based risk stratification analysis of the ISG-STS 1001 randomized trial. Cancer. 2022 Jan 1;128(1):85-93. doi: 10.1002/cncr.33895. Epub 2021 Oct 13. Erratum in: Cancer.)

Reviewer 2 Report

Comments and Suggestions for Authors

In this  review, we explore the role of non-surgical treatments in managing STS 

The topic is interesting, but the review is confusing.

Please acknowledge the narrative nature of the review both in the title and in the abstract.

The Authors, included also ALT and dermatofibrosarcoma, considered as STS. This might be misleading. 

Fig 1 unuseful.

Other entities such as desmoid fibromatosis and schwannoma are wrongly considered as STS. 

Thus, I strongly believe that the review can be misleading.  It could be better focused on adjuvant tretaments of STS, even though already well described in the Literature.

Finally, the conclusions are not supported by results. This is not a "comprehensive review". 

Comments on the Quality of English Language

Many grammar and syntax errors

Reviewer 3 Report

Comments and Suggestions for Authors

In the present work Samá et al., reviewed the literature in order to provide an overview on the non-surgical approaches in soft tissue sarcoma in different clinical scenarios. The goal of their review was to elucidate the current knowledge on these topics and stimulate further research to improve tumor control and patients’ survival. The authors have presented very well and in a way that it is easy to understand. As the authors state, “even if surgery represents the main potentially curative treatment in soft tissue sarcoma, non-surgical strategies must be considered, especially in specific clinical settings in which limitations emerge”, this is an interesting statement, on which the authors should elaborate a bit more. What is the reason, if any, that non-surgical treatments were not considered up-to-date? What is the current situation? In addition, the authors could mention recent advances, if any, in treatment (for example, vaccines, exosomes, chemotherapy etc.)

Overall, this is an interesting work, although its main focus is on the clinical setting of soft tissue sarcomas, it has merit for publication.

Comments on the Quality of English Language

minor

Reviewer 4 Report

Comments and Suggestions for Authors

The authors described the soft tissue sarcoma treatmenr. The review focused on different approaches to therapy. 
The authors must rewrte this rewiev. 
1 the sarcoma ate not carcinoma !! The first sentence in the abstract dicvalified the next all. 
Schwanoma is not sarcoma annd carcinoma too. The malignant schwanoma only rare present malignant type of tumor

desmoid is not sarcoma

fibromarosis are the binigne variant of fibtous tumors and stating lije fiboroma fibromarosis through desmoids to fibrosarcoma variant. 

the authors ignored histopathology , grading of sarcomas.

the therapy depend on histology, size ´, immunohistoloogy, location, spread and any histoloogy specific signs.

the review is confused and didnot respect the instrctions for authors.